# Human IFITM3 restricts chikungunya virus and Mayaro virus infection and is susceptible to virus-mediated counteraction

Sergej Franz[1,2], Fabian Pott[3,4] , Thomas Zillinger[5] , Christiane Schüler[3,4] , Sandra Dapa[1], Carlo Fischer[3] , Vânia Passos[1], Saskia Stenzel[3,4], Fangfang Chen[6] , Katinka Döhner[7], Gunther Hartmann[3], Beate Sodeik[7,8] , Frank Pessler[6], Graham Simmons[2], Jan Felix Drexler[3], Christine Goffinet[1,3,4]

Interferon-induced transmembrane (IFITM) proteins restrict membrane fusion and virion internalization of several enveloped viruses. The role of IFITM proteins during alphaviral infection of human cells and viral counteraction strategies are insufficiently understood. Here, we characterized the impact of human IFITMs on the entry and spread of chikungunya virus and Mayaro virus and provide first evidence for a CHIKV-mediated antagonism of IFITMs. IFITM1, 2, and 3 restricted infection at the level of alphavirus glycoprotein-mediated entry, both in the context of direct infection and cell-to-cell transmission. Relocalization of normally endosomal IFITM3 to the plasma membrane resulted in loss of antiviral activity. rs12252-C, a naturally occurring variant of *IFITM3* that may associate with severe influenza in humans, restricted CHIKV, MAYV, and influenza A virus infection as efficiently as wild-type *IFITM3*. Antivirally active IFITM variants displayed reduced cell surface levels in CHIKV-infected cells involving a posttranscriptional process mediated by one or several nonstructural protein(s) of CHIKV. Finally, IFITM3-imposed reduction of specific infectivity of nascent particles provides a rationale for the necessity of a virus-encoded counteraction strategy against this restriction factor.

## Introduction

Infection of humans by mosquito-transmitted chikungunya virus (CHIKV) and Mayaro virus (MAYV) causes, in most of the infected individuals, an acute febrile illness accompanied by symmetric joint pain and inflammation. In a subset of infected individuals of varying size depending on the outbreak, long-term morbidity manifesting as chronic arthritis with debilitating pain has been reported for both members of the Semliki Forest Virus complex and is causing growing medical concern and socioeconomical loss in affected countries. Although likely being endemic in East Africa since several centuries, chikungunya virus outbreaks of increasing frequency have occurred worldwide in the past 15 yr. MAYV is causing increasing attention in the neotropics. Neither approved vaccines nor antivirals are available against either alphavirus infection. Despite increasing importance of emerging arthritogenic alphaviruses for the human population, the biology of their replication cycle and their interplay with host proteins only begin to be elucidated. Specifically, the entry process of CHIKV and MAYV is greatly facilitated in the presence of target cell–expressed adhesion molecule MXRA8 (Zhang et al, 2018), and FHL1 serves as a cofactor for CHIKV, but not MAYV, RNA replication (Meertens et al, 2019).

Interferon-induced transmembrane (IFITM) proteins are broadly active against numerous enveloped RNA viruses, including HIV-1 (Lu et al, 2011), West Nile virus (Gorman et al, 2016), Zika virus (Savidis et al, 2016), and influenza A virus (IAV) (Brass et al, 2009), as well as enveloped DNA viruses (Li et al, 2018, 2019a). Among the five human *IFITM* genes, only *IFITM1*, *IFITM2*, and *IFITM3* have antiviral properties by restricting virus entry (Bailey et al, 2014). *IFITM* genes encode small transmembrane proteins of debated topology (Liao et al, 2019). IFITM2 and IFITM3 predominantly localize in endosomal membranes, whereas IFITM1 resides on the cell surface (Chesarino et al, 2014; Weston et al, 2014; Narayana et al, 2015; Compton et al, 2016). The mechanisms by which IFITM proteins inhibit viral infections appear to involve interference with fusion of viral and cellular membranes, resulting in virions trapped at the hemi-fusion

[1]Institute of Experimental Virology, TWINCORE Centre for Experimental and Clinical Infection Research, a Joint Venture Between the Hannover Medical School (MHH) and the Helmholtz Centre for Infection Research (HZI), Hannover, Germany [2]Vitalant Research Institute, San Francisco, CA, USA [3]Charité–Universitätsmedizin Berlin, Corporate Member of Freie Universität Berlin and Humboldt-Universität zu Berlin, Institute of Virology, Berlin, Germany [4]Berlin Institute of Health at Charité–Universitätsmedizin Berlin, Berlin, Germany [5]Institute of Clinical Chemistry and Clinical Pharmacology, University Hospital, Venusberg-Campus 1, Bonn, Germany [6]Research Group Biomarkers for Infectious Diseases, TWINCORE, Centre for Experimental and Clinical Infection Research, a Joint Venture Between the Hanover Medical School (MHH) and the Helmholtz Centre for Infection Research (HZI), Hannover, Germany [7]Institute of Virology, Hannover Medical School, Hanover, Germany [8]Cluster of Excellence RESIST (EXC 2155), Hannover Medical School, Hannover, Germany

Correspondence: christine.goffinet@charite.de

stage (Li et al, 2013; Desai et al, 2014). IFITM3 retains viral particles in late endosomes and targets them for lysosomal degradation (Feeley et al, 2011; Spence et al, 2019; Suddala et al, 2019). Experimental retargeting of IFITM3 to the cell surface, which can be induced by disruption of its Yxxθ type endocytosis motif through introduction of the $Y^{20}A$ mutation (Williams et al, 2014) or deletion of the 21 N-terminal amino acids (Weidner et al, 2010; Jia et al, 2012), nullifies its activity against many viruses which enter by endocytosis, including IAV (John et al, 2013).

The potential importance of human IFITM3 protein as antiviral factor has been addressed in clinical observation studies of influenza A. Specifically, the single-nucleotide polymorphism (SNP) rs12252-C allele might associate with increased influenza A mortality and morbidity (Everitt et al, 2012; Williams et al, 2014; Pan et al, 2017), although other cohorts failed to display this association (Mills et al, 2014; Lopez-Rodriguez et al, 2016; Randolph et al, 2017; Carter et al, 2018; David et al, 2018). A debated mechanistic working model based on the idea that the genetic variation induces alteration of a splice acceptor site, resulting in expression of a truncated IFITM3 protein which lost its antiviral activity (Everitt et al, 2012).

The impact of human IFITM proteins on alphaviral infection remains poorly elucidated, and no information on the anti-alphavirus ability of the SNP rs12252-C allele is available. In *Mus musculus*, a non-natural host of alphaviruses from the Semliki Forest Virus complex, IFITM3 shares 65% homology in amino acid sequence with the human ortholog and inhibits multiple arthritogenic alphaviruses, including CHIKV, and encephalitogenic alphaviruses in vivo (Poddar et al, 2016). Heterologously expressed human IFITM3, and to a lesser extent IFITM2 restrict Sindbis virus and Semliki Forest virus through inhibition of fusion of viral and cellular membranes (Weston et al, 2016). Furthermore, human IFITM1, 2, and 3 emerged as potential inhibitors of CHIKV infection in a high-throughput ISG overexpression screen (Schoggins et al, 2011), but their anti-alphaviral properties have not been characterized in the context of CHIKV and MAYV infection.

Here, we addressed the activity of human IFITM1, 2, and 3 as well as of naturally occurring and experimental human IFITM3 variants against CHIKV and MAYV. All three IFITM proteins restricted infection at the level of alphaviral glycoprotein-mediated entry. Experimentally induced relocalization of normally endosomal IFITM3 to the plasma membrane resulted in the loss of its antiviral activity despite robust expression. The rs12252-C allele restricted CHIKV, MAYV, and IAV as efficiently as wild-type IFITM3. Finally, steady-state IFITM3 expression levels were markedly reduced at the posttranscriptional level in productively infected cells, suggesting the existence of a so far unappreciated virus-mediated counteraction strategy of IFITM factors mediated by expression of one or several nonstructural proteins. Antagonism of IFITMs may contribute to mitigation of the potential ability of IFITM3 to negatively imprint nascent virions.

# Results

## Endogenous human IFITM3 restricts CHIKV infection

To investigate the anti-CHIKV activities of human IFITM proteins, we applied a CRISPR/Cas9-assisted approach to edit the *IFITM3* gene in

CHIKV-susceptible, IFITM3-expressing HeLa cells (Fig S1). Specifically, we functionally ablated the *IFITM3* gene by introducing a frameshift after nucleotide 84 (KO) or by deleting a large part of exon 1 of *IFITM3* (Δexon1) in both alleles. In addition, we introduced a T-to-C transition at position 89 in both *IFITM3* alleles to express the minor C allele of the SNP rs12252 (rs12252-C), which has been suggested to associate with severe IAV infections (Everitt et al, 2012; Zhang et al, 2013; Pan et al, 2017). Furthermore, we deleted a region of 31 base-pairs encompassing the primary ATG codon (Δ1st ATG) (Fig S1). In all clones, Sanger sequencing confirmed that *IFITM3* (Table 1), but not the highly homologous *IFITM2* gene (data not shown) had been edited. Immunoblotting showed that although all cell lines and clones clearly up-regulated expression of IFITM1 and the prototypic ISG product ISG15 upon IFN stimulation, IFITM3 protein detection was abrogated in HeLa cells encoding *IFITM3* KO and *IFITM3* Δexon1 (Fig 1A). In addition, no signal was detectable for IFITM3 (Δ1st ATG), arguing against expression of a truncated protein and rather for absence of expression (Fig 1A). In contrast, three clones expressing the rs12252 T-to-C variant displayed detectable IFN-induced IFITM3 expression, although slightly lower than in parental cells. Importantly, the immunoblot provided no evidence for expression of a truncated IFITM3 protein in rs12252-C cells, but rather displayed a band of equal molecular weight as the one from wild-type IFITM3 (Fig 1A). Detection of up-regulated IFITM2 protein failed despite the confirmed specificity of the IFITM2-targeting antibody (see Fig 2A). Flow cytometry analysis of IFITM3 protein expression in permeabilized cells paralleled the results of the immunoblot analysis, with IFITM3 in rs12252-C cells being slightly less abundant at baseline and upon IFN stimulation (Fig 1B). Finally, immunofluorescence microscopy confirmed expression of IFITM3 protein in parental and in rs12252-C cells, but not in the other cell lines (Fig 1C).

We then inoculated the individual cell lines with CHIKV. Throughout the study, we used the Indian Ocean lineage strain LR2006 harboring an EGFP reporter at the 5′ end of the viral genome, hereafter referred to as EGFP-CHIKV. Absence of IFITM3 expression in the KO, Δexon1, and $Δ^{1st}$ ATG cells was accompanied by diminished effectivity of the IFN-induced antiviral program, whereas IFITM3 expression in parental and rs12252-C cells impaired EGFP-CHIKV infection to similar levels (Figs 1D and S2). Similarly, the susceptibility of $Δ^{1st}$ ATG cells to EGFP-MAYV (Li et al, 2019b) infection was markedly enhanced (Figs 1E and S2). Finally, parental cells and rs12252-C cells were equally restrictive to IAV infection, as opposed to KO cells (Fig 1F). In conclusion, we show that endogenous IFITM3 restricts CHIKV and MAYV infection. Furthermore, the naturally occurring variant of *IFITM3*, rs12252-C, drove expression of an IFITM3 protein that displays a similar anti-CHIKV and anti-IAV potency as the major *IFITM3* allele.

## Ectopic expression of human IFITM-HA proteins restricts alphaviral infection by interfering with glycoprotein-mediated entry

To gain mechanistic insight into the differential antiviral potency of IFITM proteins and variants, we stably transduced HEK293T cells, which lack endogenous IFITM protein expression, with C-terminally HA-tagged IFITM1, 2, or 3 (Brass et al, 2009). In addition, we generated individual cell lines expressing the mutant proteins

**Table 1.** List of *IFITM3* variants generated by gene editing of HeLa cells

| *IFITM3* clone | Type of gene edit | Gene edit (nt position refers to NM_021034.3) | Genomic DNA sequence | Guide RNA target (PAM) |
|---|---|---|---|---|
| *KO* | +1 frameshift | Insertion of A between nts 84 and 85 | CCTGTCA[**+A**]ACAGTGGCCAG | GGGGGCTGGCCACTGTTGAC(AGG) |
| *Δexon1* | Deletion | Deletion of 381 nts post nt 36 | CGACCGCCGCTGGTCTT [deletion of 381 nt] | 1:CGACCGCCGCTGGTCTTCGC(TGG) |
| | | | TCCCGTGTGTGCCCACG | 2:CGTGGGCACACACGGGACAG(AGG) |
| *SNP rs12252-C* | T to C transition | T89C | CCTGTCAACAG**C**GGCCAGCCCCC | guide:GGGGGCTGGCCACTGTTGAC(AGG) |
| | | | | +HDR-Donor: CTGGACACCATGAATCACACTGTCCAAACCTTCTTCTCTCCTGT |
| | | | | CAACAGCGGCCAGCCCCCCAACTATGAGATGCTCAAGGAGGAGCAC |
| *Δ1st ATG* | Deletion | Deletion of 31 nts post nt 36 | CGACCGCCGCTGGTCTT [deletion of 31 nt] | 1:CGACCGCCGCTGGTCTTCGC(TGG) |
| | | | CTTCTTCTCTCCTGTCAA | 2:TGACAGGAGAGAAGAAGGTT(TGG) |

nt, nucleotide.

IFITM3($Y^{20}$A)-HA and IFITM3(Δ1-21)-HA, which localize to the plasma membrane (Weidner et al, 2010; Feeley et al, 2011; Jia et al, 2012; Williams et al, 2014). Finally, we overexpressed the IFITM3 rs12252-C variant (IFITM3[rs12252-C]-HA). Appropriate protein expression was confirmed by immunoblot analysis using both an anti-HA antibody and primary antibodies targeting the individual authentic IFITM proteins (Fig 2A). We confirmed similar expression levels of all proteins and variants by flow cytometry (Fig 2B) and immunofluorescence microscopy (Fig 2C) using an anti-HA antibody in intact and in permeabilized cells. Thereby, we confirmed the previously reported, predominant cell surface localization of IFITM3($Y^{20}$A)-HA and IFITM3(Δ1-21)-HA (Perreira et al, 2013). IFITM3(rs12252-C)-HA did not express a shorter variant and the protein localized predominantly intracellularly, as observed in HeLa cells (Fig 1). To assess the antiviral capacity of the IFITM-HA proteins, we infected the cell lines with EGFP-CHIKV and determined the amount of EGFP-positive cells at 24 h postinfection. Over a wide range of MOIs, IFITM1-HA and IFITM3-HA, and to a lesser extent IFITM2-HA, displayed antiviral activity against CHIKV (Figs 2D and S3A and B). IFITM3(rs12252-C)-HA restricted EGFP-CHIKV infection at a similar efficiency as IFITM3-HA. In contrast, IFITM3($Y^{20}$A)-HA and IFITM3(Δ1-21)-HA failed to impair infection (Figs 2D and S3A and B). We observed a similar pattern of inhibition at the level of infectivity released in the culture supernatant of EGFP-CHIKV- and MAYV-infected cells, as judged by plaque assays (Fig 2E). To monitor whether inhibition of CHIKV infection was related to the established ability of IFITMs to interfere with enveloped virus entry, we quantified the cells' susceptibility to transduction by lentiviral pseudoparticles decorated with heterologous viral glycoproteins and expressing a luciferase cassette (Fig 2F). In line with previous reports (Brass et al, 2009), IFITM protein expression did not reduce the susceptibility of cells to transduction by particles pseudotyped with glycoproteins of murine leukemia virus. In contrast, expression of IFITM2-HA and IFITM3-HA, but not IFITM1-HA, reduced the cells' susceptibility to transduction by Ebola virus glycoprotein-pseudotyped particles, as reported before (Brass et al, 2009; Wrensch et al, 2015). Transduction of cells by particles incorporating CHIKV glycoproteins was impaired by IFITM1-HA and IFITM3-HA and, to a lesser extent, by IFITM2-HA. The

$A^{226}$V mutation in CHIKV E1 that enabled adaptation of the virus to the alternative vector *Aedes albopictus* (Tsetsarkin et al, 2007) did not overtly modulate susceptibility to IFITM protein-mediated restriction (Fig 2F). CHIKV glycoproteins from sublineages 1–4, which are derived from the S27 strain (Tsetsarkin et al, 2014), shared susceptibility to IFITM-mediated inhibition, although to slightly different extents (Fig 2F). Collectively, these data establish that IFITM proteins restrict CHIKV and MAYV, and that inhibition is directed against entry mediated by glycoproteins of different CHIKV sublineages. In addition, antiviral activity is only exerted by IFITM3 variants with reported endosomal localization.

### Cell-to-cell transmission of and direct infection by CHIKV differ in efficiency, but share susceptibility to IFITM-mediated restriction

Cell-to-cell transmission of virions can antagonize or evade the IFN-induced antiviral state, including restriction by some (Richardson et al, 2008; Vendrame et al, 2009; Jolly et al, 2010), but not all (Puigdomenech et al, 2013) antiviral factors. Expression of IFITM3 renders target cells less susceptible to infection by cell-free HIV-1 particles, but cell-to-cell transmission of HIV-1 partially escapes this antiviral effect (Compton et al, 2014). We thus addressed the relative ability of cell-associated and cell-free transmission modes in supporting CHIKV spread, and their relative susceptibility to IFITM protein-mediated restriction. As a surrogate system for cell-to-cell transmission, we subjected infected cells with an agarose overlay, which favors transfer of virions between neighboring cells and impairs infection by freely diffusing virus particles. In this set-up, the initial infection is seeded by cell-free virus, whereas subsequent rounds of infection are mediated by cell-associated virus and cell-to-cell spread. For comparison, infected cells were cultured in regular growth medium without agarose. EGFP-CHIKV spread was enhanced under agarose overlay conditions, with to twofold higher percentages of EGFP-positive cells as compared to cells cultured in regular medium (Fig 3A). Of note, in cells under agarose, an EGFP-encoding HSV-1 spread at identical or even lower efficiencies than in regular medium (Figs 3A and S4A). Fluorescence microscopy analysis of EGFP-CHIKV-infected

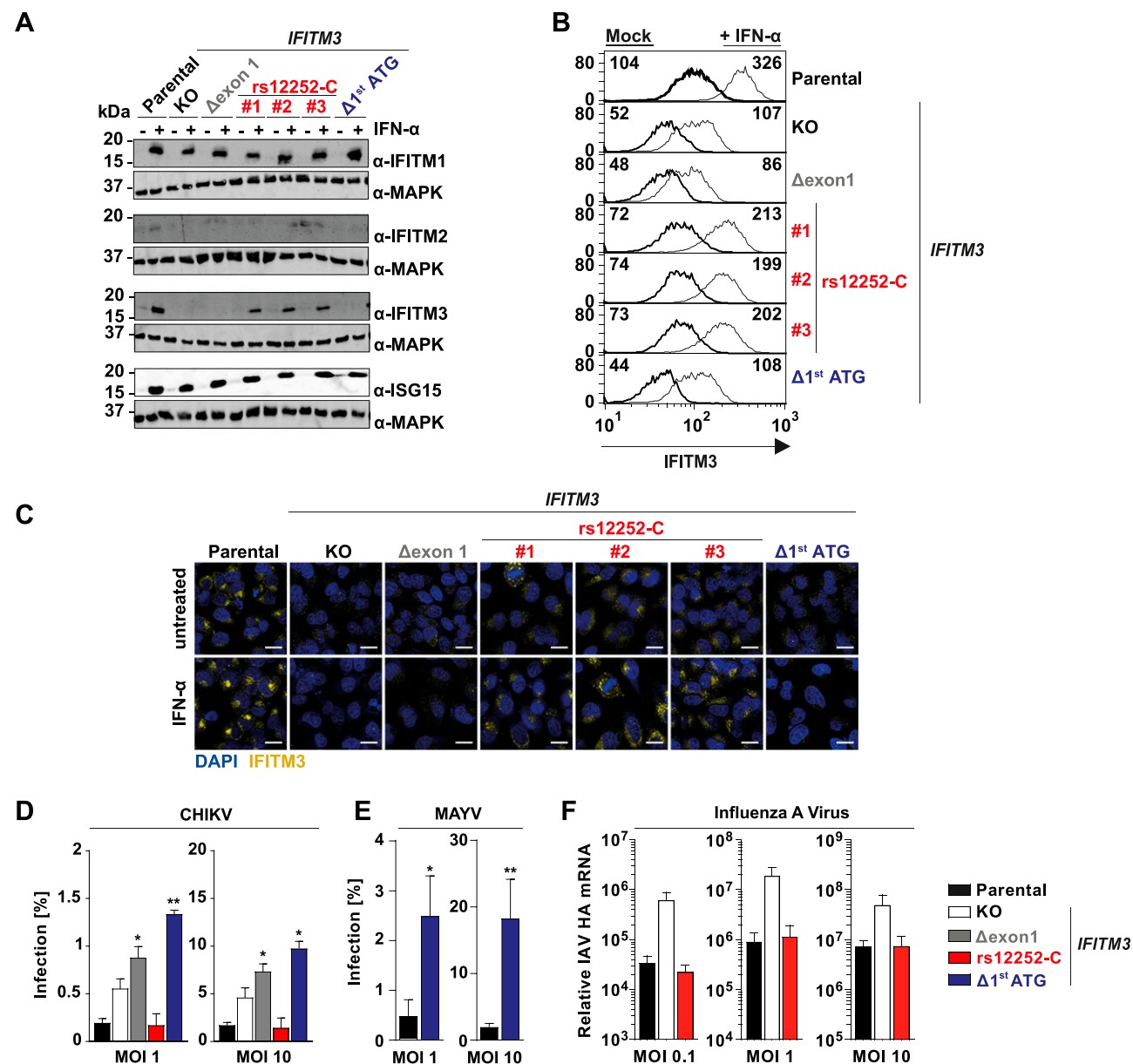

**Figure 1. Endogenous human IFITM3 restricts CHIKV infection.**
**(A)** Immunoblot of indicated HeLa cell lysates using indicated antibodies. **(B)** Flow cytometry analysis of interferon-induced transmembrane expression in permeabilized HeLa cells. **(C)** Immunofluorescence microscopy analysis of indicated HeLa cell lines/clones upon mock treatment and treatment with 5,000 IU/ml IFN-α for 48 h (scale bar = 20 μm). **(D)** HeLa cells were infected with EGFP-CHIKV after a 6 h IFN-α pre-treatment duration at indicated MOIs. The percentage of EGFP-positive infected cells was quantified 24 h postinfection by flow cytometry. **(E)** HeLa cells were infected with MAYV-GFP at indicated MOIs The percentage of EGFP-positive infected cells was quantified 24 h postinfection by flow cytometry. **(F)** HeLa cells were infected with Influenza A virus at indicated MOIs. 24 h postinfection, cell-associated viral HA mRNA was quantified by quantitative RT-PCR. SNP, single nucleotide polymorphism; MFI, mean fluorescence intensity; MOI, multiplicity of infection. Source data are available for this figure.

cells cultured under agarose overlay revealed a nest-like pattern, reflecting spread of one initial infection event to closely surrounding cells through intercellular contacts, in contrast to the more evenly distributed, punctate pattern consistent with spread of the infection by freely diffusing virus particles (Fig 3B).

We next investigated the restriction potential of the individual IFITM proteins and variants in the context of cell-free and cell-associated CHIKV infection (Figs 3C and S4B). For any cell line, the percentage of EGFP-positive cells was consistently elevated under agarose overlay conditions as compared with culture in regular medium. The fold increase was similar in all cell lines, indicating that agarose overlay treatment yielded a better spread irrespective of expression of IFITM proteins. Importantly, CHIKV was unable to escape IFITM-mediated restriction by cell-to-cell transmission (Figs 3C and S4B). Taken together, CHIKV spreads efficiently through cell-to-cell transmission, surpassing the spreading efficiency in a

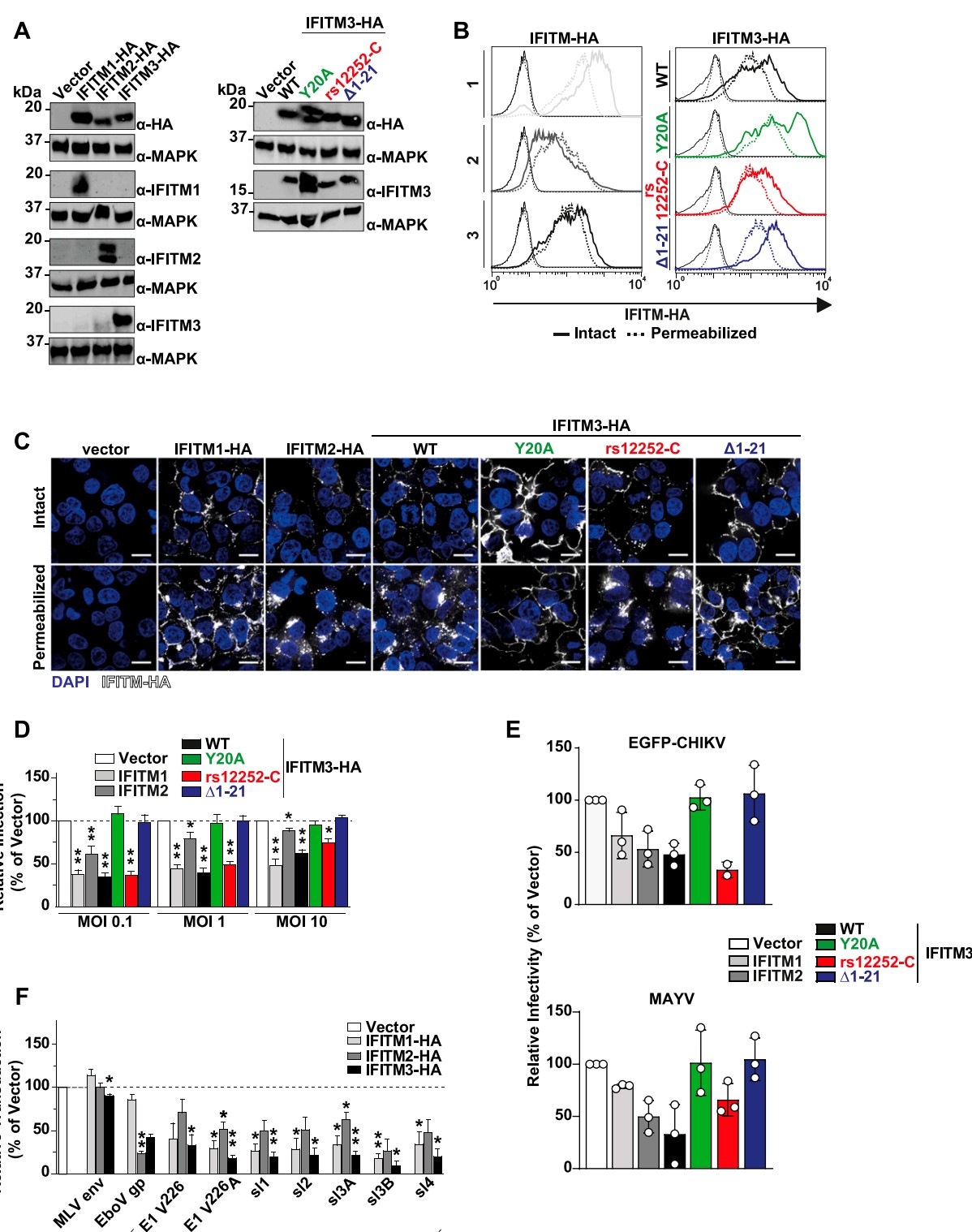

**Figure 2.  Ectopic expression of human interferon-induced transmembrane (IFITM)-HA proteins restricts alphaviral infection by interfering with glycoprotein-mediated entry.**

HEK293T cell lines stably expressing HA-tagged IFITM proteins were generated via retroviral transduction and puromycin selection. **(A)** Protein expression was evaluated by immunoblotting using indicated antibodies. **(B)** Cells were immunostained with an anti-HA antibody and analyzed by flow cytometry, measuring IFITM surface levels in membrane-intact cells and total expression in permeabilized cells. Mean fluorescence intensity is inset for each condition. **(C)** Immunofluorescence microscopy of surface and intracellular immunostaining of heterologously expressed IFITM proteins using anti-HA antibody (scale bar = 20 μm). **(D)** IFITM-HA expressing cells were

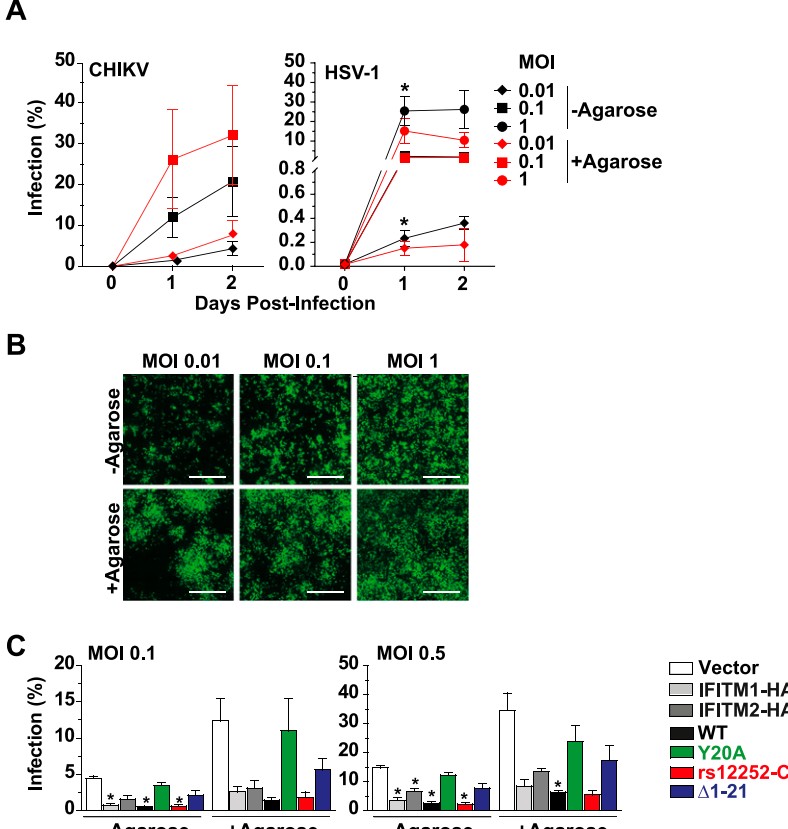

**Figure 3. Cell-to-cell transmission of and direct infection by CHIKV differ in efficiency, but share susceptibility to interferon-induced transmembrane-mediated restriction.**

Confluent HEK293T cells were infected with EGFP-CHIKV or EGFP-HSV-1 for 6 h. Subsequently, cells were overlaid with 0.8% agarose or left untreated. **(A, B)** EGFP-positive cells were measured 1 and 2 d postinfection via flow cytometry and (B) microscopic analysis of CHIKV-infected cells was performed 24 h postinfection (scale bar = 500 μm). **(C)** Confluent HEK293T cells ectopically expressing interferon-induced transmembrane-HA proteins were infected with EGFP-CHIKV for 6 h followed by the application of a 0.8% agarose overlay where indicated. 24 h postinfection, EGFP-positive cells were measured via flow cytometry.

regular culture in which virions diffuse freely. However, the restriction pattern of IFITM proteins and variants is maintained in the context of cell-to-cell transmission.

### Reduction of cell surface expression of antiviral IFITM proteins in CHIKV-infected cells

Flow cytometric analysis of CHIKV-infected cells showed a markedly lowered cell surface abundance of most, but not all IFITM-HA proteins in productively infected, EGFP-positive cells as opposed to bystander, EGFP-negative cells of the identical culture (Fig 4A). The decrease of cell surface levels was most pronounced for IFITM3-HA and IFITM-3(rs12252-C)-HA, followed by IFITM2-HA. Surface levels of IFITM1-HA, IFITM3(Y20A)-HA, and IFITM3(Δ1-21)-HA, which share a predominantly cell surface localization, were clearly less affected (Fig 4A and B). Whereas quantitative immunoblotting of whole cell lysates failed to detect a reduction of IFITMs upon infection (Fig 4C), single-cell inspection of EGFP-positive cells by immunofluorescence suggested that antivirally active IFITMs were reduced in quantity upon infection (Fig 4D). To address if reduction

of selected IFITM-HA protein expression in EGFP-positive cells requires the entry process of CHIKV infection, we bypassed the entry step by directly transfecting the full-length CHIKV RNA genome. Strikingly, IFITM-HA proteins were also down-regulated in EGFP-CHIKV-transfected cells, in stark contrast to cells transfected with HIV-1 EGFP DNA (Figs 4E and S5). Furthermore, HEK293T IFITM3-HA cells transfected with either EGFP-CHIKV genomic RNA or mRNA encoding CHIKV nsP1-4 and EGFP shared reduction of IFITM3-HA levels on their surface specifically in EGFP-positive cells (Fig 4F). These data suggest that expression of one or several nonstructural CHIKV proteins induces the down-regulation of antivirally active IFITM proteins. In contrast, inactive IFITM3 variants that localize predominantly at the cell surface are less susceptible to this down-regulation.

### Posttranscriptional reduction of endogenous IFITM3 expression in CHIKV- and MAYV-infected cells

Next, we analyzed the consequence of CHIKV infection on endogenous IFITM3 expression levels. By immunofluorescence microscopy,

infected with EGFP-CHIKV of increasing MOIs for 24 h, and the percentage of EGFP-positive cells was quantified by flow cytometry. **(E)** IFITM-HA expressing cells were infected with EGFP-CHIKV or MAYV strain TRVL15537. 24 h postinfection, supernatant was collected and plaque assays were performed on Vero E6 cells to quantify infectious virus progeny. **(F)** Cell lines were inoculated for 48 h with lentiviral firefly luciferase-expressing reporter pseudoparticles decorated with glycoproteins of murine leukemia virus, Ebola virus, or different CHIKV lineages. Luciferase activity in transduced cells was quantified luminometrically and normalized to the vector control cell line.

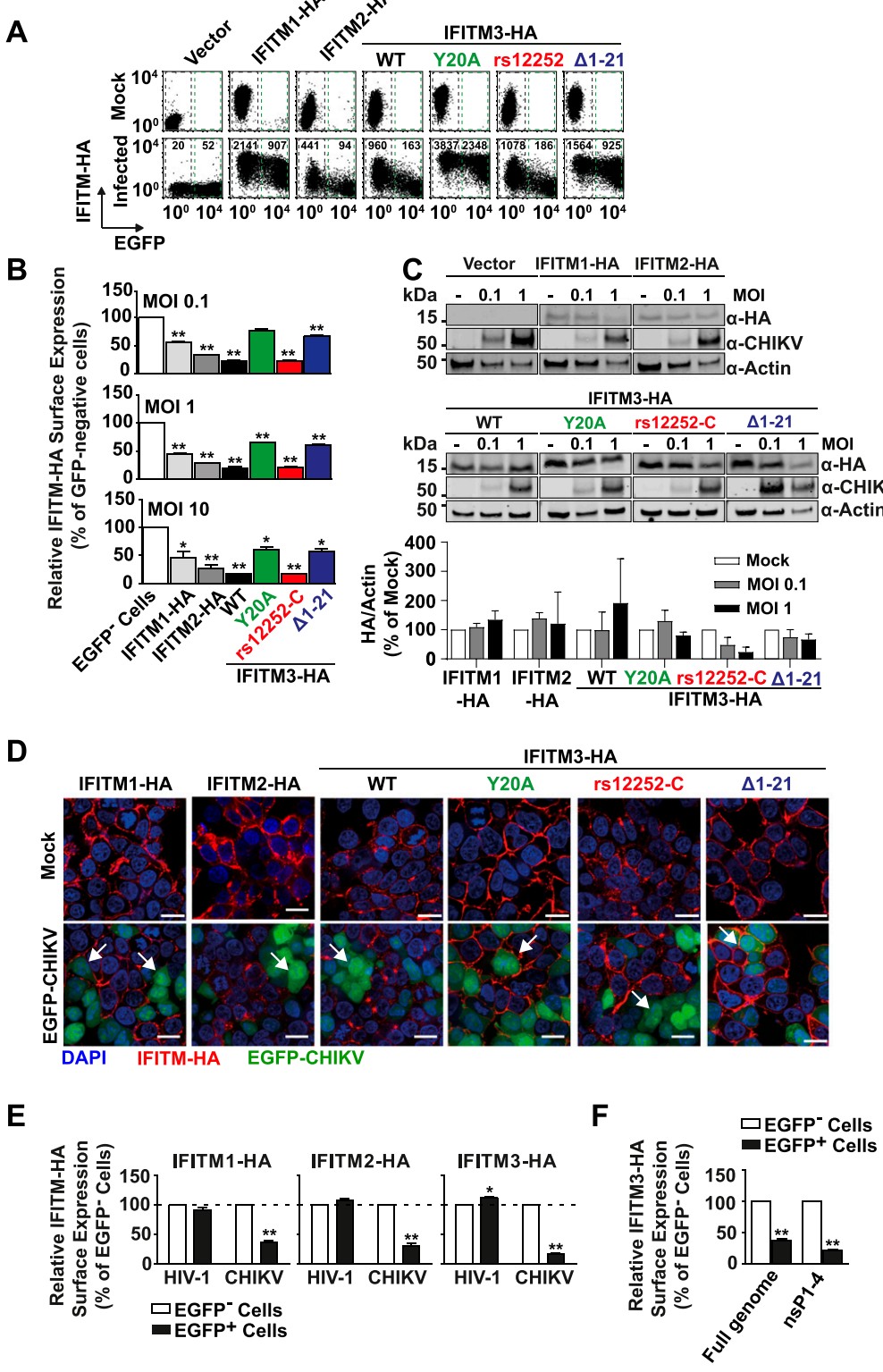

**Figure 4. Reduction of cell surface expression of antiviral interferon-induced transmembrane (IFITM) proteins in CHIKV-infected cells.**
**(A)** Indicated HEK293T cells were infected with EGFP-CHIKV (MOI 10) for 24 h and were immunostained with an anti-HA antibody for flow cytometry. **(B)** Quantification of IFITM-HA expression in EGFP-positive relative to EGFP-negative cells after infection with EGFP-CHIKV for 24 h. **(C)** Immunoblot analysis of IFITM-HA expression in cells infected with EGFP-CHIKV at the indicated MOIs and quantitative analysis thereof. **(D)** Indicated HEK293T cell lines were infected with EGFP-CHIKV (MOI 10) for 24 h. Permeabilized cells were immunostained with an anti-HA antibody and analyzed microscopically (scale bar = 20 μm). Arrowheads indicate EGFP-positive cells with IFITM-HA expression (red) or lack thereof. **(E)** Indicated cell lines were transfected individually with full-length EGFP-CHIKV RNA and HIV-1 EGFP DNA and were immunostained with an anti-HA antibody. Mean fluorescence intensitys of IFITM-HA EGFP-positive cells were determined via flow cytometry and normalized to the mean fluorescence intensity of EGFP-negative cells. **(F)** Quantification of IFITM3-HA surface level 24 h after individual transfection of HEK293T IFITM-3-HA cells with full length EGFP-CHIKV mRNA or RNA encoding the non-structural proteins 1–4 and EGFP.
Source data are available for this figure.

we quantified the amounts of IFITM3 protein in productively infected, EGFP-positive, and bystander, EGFP-negative HeLa cells (Fig 5A). Similar to the heterologous expression set-up (Fig 4), we detected a strong reduction of endogenous IFITM3 immunostaining intensity in highly EGFP-positive cells, whereas infected cells with low EGFP intensity and EGFP-negative cells displayed similarly high levels of IFITM3 (Fig 5A and B). In stark contrast to IFITM3, ISG15, and MX1 protein levels remained similar in EGFP-positive and EGFP-negative

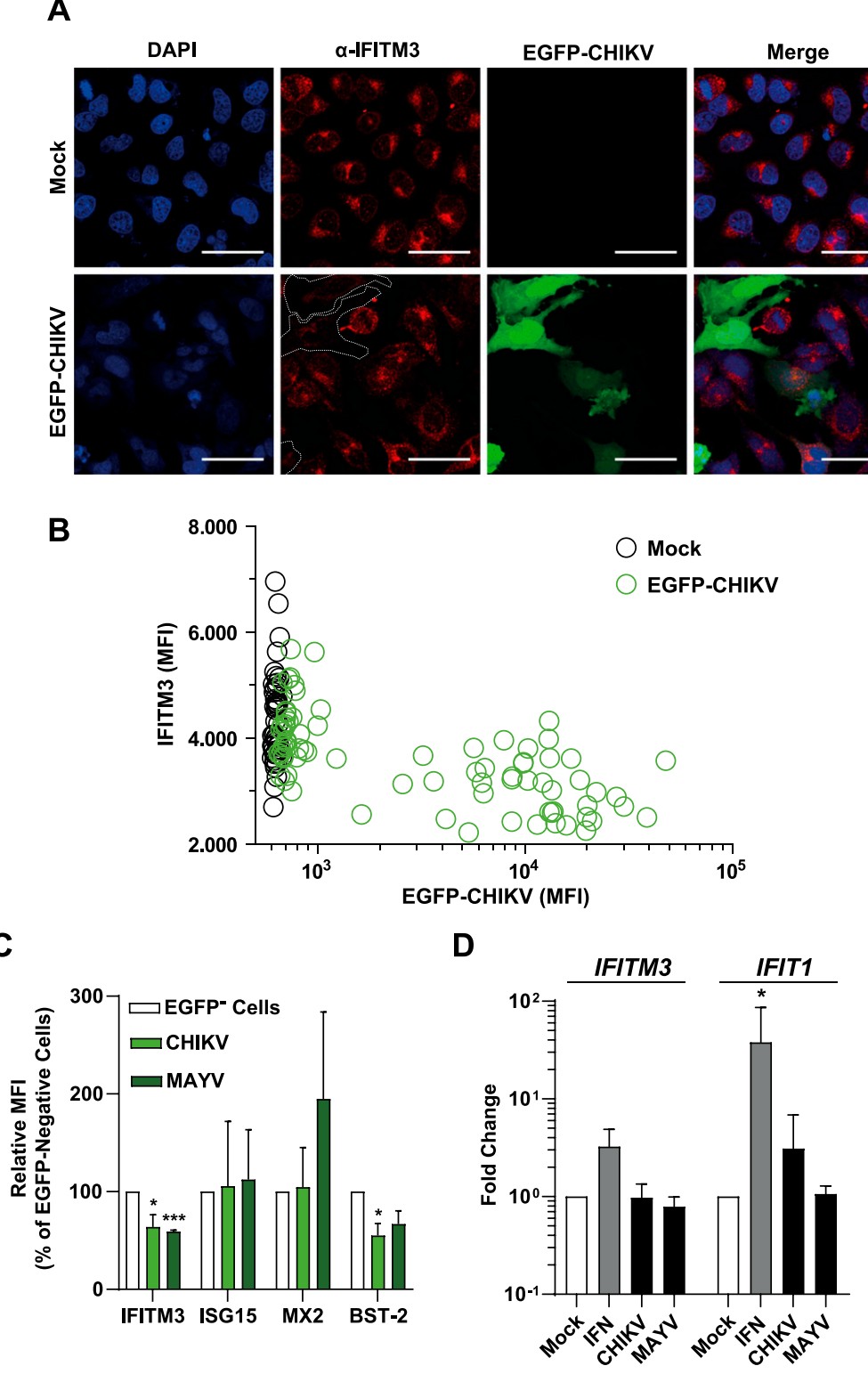

**Figure 5. Posttranscriptional reduction of endogenous IFITM3 expression in CHIKV-infected cells.**
**(A)** HeLa cells were infected with EGFP-CHIKV (MOI 100) for 24 h, permeabilized, and immunostained for IFITM3 before microscopic analysis. Dotted lines indicate the border of EGFP-positive cells (scale bar = 50 μm). **(B)** Quantification of microscopic images of infected HeLa cells. IFITM3 mean fluorescence intensity (MFI) was determined using ImageJ and is plotted against EGFP MFI (mock cells n = 50; CHIKV-infected cells n = 80). **(C)** HeLa cells were infected with EGFP-CHIKV and EGFP-MAYV (MOI 10) for 24 h. Subsequently, permeabilized cells were stained for IFITM3, ISG15, MX1, or BST-2 and MFI values were normalized to EGFP-negative cells. **(D)** HeLa cells were infected with EGFP-CHIKV or EGFP-MAYV (MOI 10) and 24 h postinfection, *IFITM3*, and *IFIT1* mRNA levels were measured by quantitative RT-PCR.
Source data are available for this figure.

cells upon CHIKV and MAYV infection (Figs 5C and S6), arguing against a global translational shut-off as the underlying reason and pointing towards a specific effect on IFITM3. Interestingly, the release inhibitor Tetherin/CD317/BST-2 (Neil et al, 2008; Van Damme et al, 2008) seemed also to be down-regulated upon CHIKV and MAYV infection, although statistical significance was reached only for CHIKV (Figs 5C and S6). Finally, CHIKV and MAYV infection failed to modulate *IFITM3* mRNA expression in HeLa cells (Fig 5D), excluding that infection

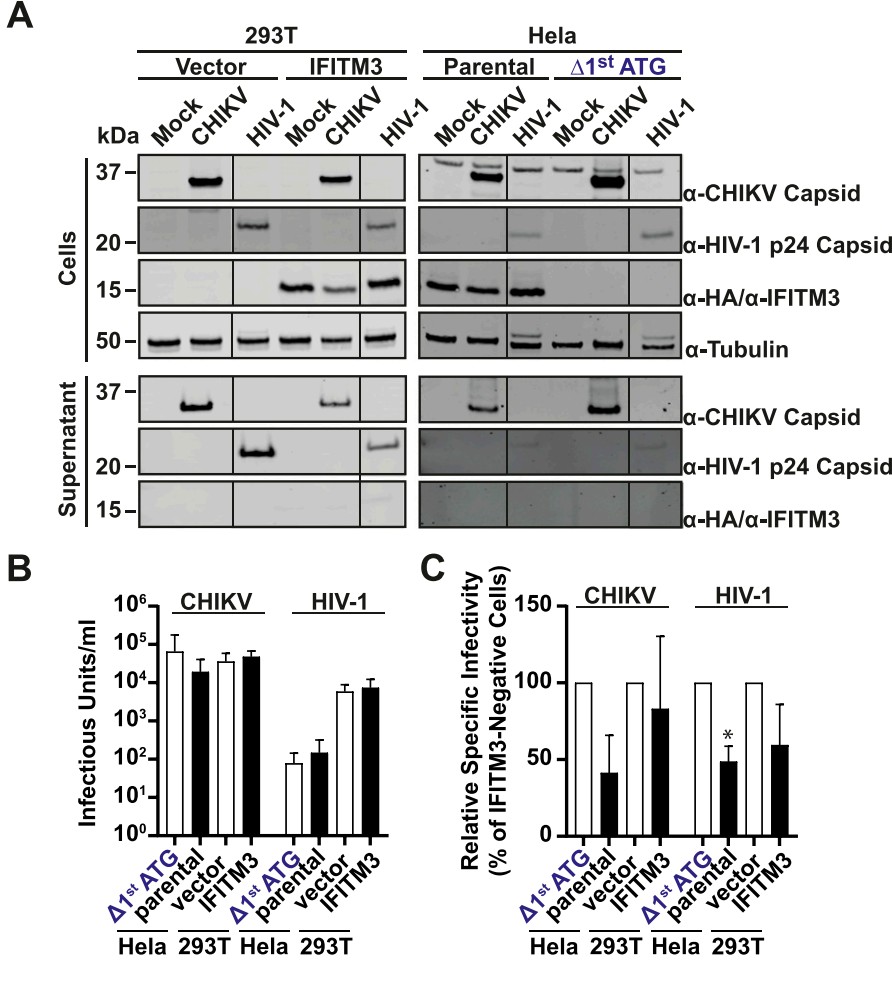

**Figure 6. Reduction of virion infectivity through expression of IFITM3 in virus-producing cells.**
**(A)** Immunoblot analysis of lysates and concentrated supernatants of 293T and Hela cells transfected with full-length EGFP-CHIKV mRNA or pBR-NL4.3-GFP plasmid at 24 h posttransfection. **(B)** Viral titers produced by cells transfected in (A) normalized to the relative amount of GFP-positive cells. Viral titers were determined by titration and flow cytometric analysis of supernatants on 293T cells (CHIKV) or Tzm-bl cells (HIV), respectively. **(C)** Relative specific infectivity of produced virions as calculated by the infectivity per abundance of viral capsid.
Source data are available for this figure.

modulates transcription of the *IFITM3* gene, and suggesting an infection-induced modulation of IFITM3 protein steady-state levels. IFN-*α* treatment boosted *IFITM3* mRNA levels, as expected. Furthermore, another prototypic ISG, *IFIT1*, was clearly induced by both IFN-*α* treatment and CHIKV infection (Fig 5D). Conclusively, these data support a posttranscriptional modulation of IFITM3 expression in CHIKV- and MAYV-infected cells, potentially representing a virus-mediated counteraction strategy of IFITM restriction.

### Reduction of particle infectivity through expression of endogenous IFITM3 in alphavirus-producing cells

Reduction of IFITM3 protein abundance upon alphaviral infection suggested that IFITM3 exerted antiviral functions beyond entry inhibition. Therefore, we determined the impact of IFITM3 expression in virus-producing cells on the infectivity of progeny virions. IFITM3 can incorporate into viral particles and reduce their ability to infect new cells (Compton et al, 2014; Tartour et al, 2014, 2017; Appourchaux et al, 2019). To bypass the impact of IFITM proteins on virus entry, we transfected cells expressing IFITM3 endogenously or exogenously, and their IFITM3-negative counterparts. 24 h posttransfection, we analyzed the abundance of viral capsid and amount of infectivity in the supernatant (Fig 6A). Capsid

abundance was lower in supernatants from IFITM3-expressing cells, which however was related to a generally lower transfection rate and lower cell-associated capsid expression. We failed to detect virion-associated IFITM3 in the supernatant of any infected cell cultures under these experimental conditions (Fig 6A). Viral titers, when normalized for differences in transfection efficiencies, were not significantly influenced by the IFITM3 expression status (Fig 6B). However, the specific infectivity, defined as the infectivity per capsid, was twofold reduced for CHIKV and HIV-1 produced in IFITM3-expressing parental HeLa cells and for HIV-1 produced in HEK293T cells expressing IFITM3-HA (Fig 6C). However, the specific infectivity of CHIKV did not seem to be affected by heterologous IFITM3 expression (Fig 6C). These data provide first evidence for a potential ability of endogenous IFITM3 to negatively imprint nascent CHIKV, in addition to inhibition of virus entry. Future work is required to establish the relative contribution of these two IFITM3-mediated antiviral strategies in the context of alphaviral infection.

## Discussion

First evidence for anti-CHIKV properties of human IFITM proteins 1, 2, and 3 was provided in a high-throughput ISG overexpression

screen in STAT-deficient human fibroblasts (Schoggins et al, 2011). However, the pattern of IFITM-mediated CHIKV inhibition and potential CHIKV-mediated counteraction strategies remain obscure. By applying a two-armed approach that included investigation of endogenous *IFITM3* and variations thereof, and of heterologously expressed, epitope-tagged IFITM proteins and mutants, we established and characterized the antiviral activity of IFITM proteins against CHIKV and MAYV.

Our study centered on the investigation of IFITM3's role on alphaviral infection. Expression of endogenous and heterologous IFITM3 rendered HeLa and HEK293T cells less susceptible to CHIKV infection, respectively. In the endogenous expression context, frameshift insertion in the 5' part of the *IFITM3* gene, deletion of a large part of the first exon of the *IFITM3* gene or of a smaller, 31-base pair region comprising the first, canonical ATG, resulted in ablation of IFITM3 expression and antiviral activity. Similarly, cells expressing a frameshift in the 5' part of the first exon of the *IFITM3* gene were unable to restrict IAV infection and lost the ability to restrict CHIKV.

It has been hypothesized that the T-to-C substitution in the SNP rs-12252, located in the first exon of the *IFITM3* gene, alters a splice acceptor site, resulting in a truncated IFITM3 protein lacking its first 21 amino acids and exerting reduced antiviral activity (Everitt et al, 2012). However, this working model did not substantiate because neither the predicted alternatively spliced mRNA (Randolph et al, 2017; Makvandi-Nejad et al, 2018) nor a truncated IFITM3 protein (Makvandi-Nejad et al, 2018) has been detected in cells homozygously expressing rs12252-C. In line with these negative results, we detect a protein of normal size both in cells homozygously expressing the SNP and in cells overexpressing an IFITM3-encoding construct carrying the T-to-C transition. However, our anti-IFITM3 immunoblotting technique is clearly able to detect the smaller molecular weight of IFITM3(Δ1-21)-HA, excluding a technical inability to detect marginally smaller IFITM3 proteins in general. Whereas IFITM3(Δ1-21)-HA localized to the cell surface and lacks antiviral activity, subcellular localization and antiviral phenotype of the rs12252-C variant remained indistinguishable from the wild-type IFITM3 in terms of anti-CHIKV, anti-MAYV, and anti-IAV activity, molecular weight and subcellular localization. However, IFITM-3 rs12252-C seemed to be expressed to slightly lower levels than wild-type IFITM3, without impacting its antiviral efficacy. Together, we conclude that if any, specific functional properties of the rs12252-C allele remain to be discovered and may not directly be implicated in cell-intrinsic immunity.

Experimental redirection of IFITM3 to the cell surface by disrupting the sorting motif YxxΦ (Jia et al, 2012, 2014) through introduction of the Y$^{20}$A or Δ1-21 mutations resulted in a loss of its anti-CHIKV and anti-MAYV activity. Analogous findings have been obtained by others for SINV and SFV (Weston et al, 2016), and other enveloped viruses which invade cells via receptor-mediated endocytosis (John et al, 2013; Jia et al, 2014). In addition, heterologous assays revealed an anti-CHIKV and anti-MAYV activity of IFITM1 that equaled that of IFITM3. This observation contrasts reports by others for Sindbis virus and Semliki Forest virus (Weston et al, 2016) but is in accordance with results obtained in a study that screened the antiviral potential of several ISGs against CHIKV (Schoggins et al, 2011). Heterologous expression of genes can cause aberrant subcellular localization and/or nonphysiological expression levels.

Therefore, results obtained by heterologous expression need to be interpreted with caution. Importantly, levels of IFITM protein expression obtained by heterologous expression of IFITM were similar to levels induced by IFN treatment of HeLa cells (data not shown). Future studies are warranted to corroborate the contribution of IFITM1 to alphavirus restriction.

Restriction was directed against CHIKV E2/E1 glycoprotein-mediated entry, was maintained in the context of CHIKV glycoproteins expressing the vector switch-enabling mutation A$^{226}$V in CHIKV E1 and displayed a significant breadth because glycoproteins of several CHIKV S27 strain sublineages were sensitive to IFITM protein-mediated restriction.

Most of the cell culture studies on CHIKV are conducted using cell-free, purified virus. An early report published in 1970 suggested that CHIKV spreads via cell-to-cell contacts when free virions are immunologically neutralized (Hahon & Zimmerman, 1970). Along this line, intercellular transmission of CHIKV was reported more recently to be less sensitive to antibody-mediated neutralization (Lee et al, 2011). In the present study, we applied a semi-solid, agarose-containing overlay to infected cultures to determine the contribution of cell-free virus versus intercellular transmission, as performed by others for hepatitis C virus (Timpe et al, 2008), vesicular stomatitis virus and murine leukemia virus (Liberatore et al, 2017). Interestingly, CHIKV tended to spread more efficiently under agarose overlay, as opposed to HSV-1. The exact mode of intercellular transmission of CHIKV in different cell types will be an important future object of investigation and might contribute to understanding alphavirus persistence in vivo. The relative restriction potential of individual IFITM proteins and variants was identical in the cell-free and cell-associated transmission set-ups. As a contrasting example, HIV-1 is able to overcome IFITM3-mediated restriction via cell-to-cell spread; however, only when IFITM3 is expressed solely on target cells (Compton et al, 2014). Remarkably, IFITM proteins display a second layer of antiviral activity, which consists in diminishing the infectivity and fusogenicity of enveloped virus particles produced in IFITM protein–expressing cells (Compton et al, 2014; Tartour et al, 2014). This process is often, but not always, accompanied by incorporation of IFITM proteins into the membrane of secreted virions (Appourchaux et al, 2019). Interestingly, our identification of a reduced particle infectivity in IFITM3-expressing HeLa cells argues for the ability of IFITM3 to negatively imprint CHIKV and MAYV. Interestingly, this phenotype was absent in HEK293T cells heterologously expressing IFITM3, suggesting differences depending on the expression context and quantities, which have already been reported in the context of other viral infections (Bozzo et al, 2020 *Preprint*).

CHIKV has evolved several strategies to evade or antagonize cell-intrinsic immunity, facilitating successful replication and spread in its host. The multifunctional nonstructural protein nsP2 of CHIKV, besides proteolytically processing the nonstructural polyprotein precursor (Rausalu et al, 2016), inhibits IFN-induced nuclear translocation of STAT1 in Vero cell lines (Fros et al, 2010). Furthermore, it degrades the Rpb1 subunit of the RNA polymerase II in BHK-21 and NIH3T3, but not mosquito cells (Akhrymuk et al, 2012). In addition, translational shutoff has been observed in some CHIKV-infected cell lines (White et al, 2011). Here, in two different cellular systems, we obtained no evidence for a broad

transcriptional or translational shutoff indicating some cell line or cell type specificity. On the contrary, productive CHIKV infection associates with reduced IFITM3 protein levels, and this reduction appears to operate at the posttranscriptional level. In the HEK293T-based heterologous expression system, only antivirally active, endosomally located (WT; rs-12252-C) but not inactive, plasma membrane–resident ($Y^{20}A$; Δ1-21) IFITM3 proteins were reduced in quantity in CHIKV-infected cells. This specificity was observed despite all IFITM-HA proteins being heterologously expressed under the control of the identical CMV immediate early promoter, which is unlikely to be targeted by CHIKV evasion strategies. In CHIKV-infected HeLa cells, endogenous IFITM3 protein levels were reduced in the absence of a detectable net decrease of *IFITM3* mRNA, again pointing towards a specific counteraction strategy directed against the IFITM3 protein. Interestingly, IFITM3 degradation has been reported to occur through ubiquitination of a highly conserved PPxY motif that overlaps with the aforementioned YxxΦ endocytosis motif by the E3 ubiquitin ligase NEDD4 (Chesarino et al, 2015). With $Y^{20}$ representing both a critical phosphorylation site required for IFITM3 internalization and part of a ubiquitination motif important for degradation of the protein (Yount et al, 2012; Chesarino et al, 2014), it is tempting to speculate that a nonstructural protein of CHIKV promotes, either directly or indirectly, endocytosis and/or ubiquitination-dependent degradation of IFITM3, a process that has yet to be studied in detail.

# Materials and Methods

### Cell lines

BHK-21, HeLa (CCL-2; ATCC), HEK293T (CRL-3216; ATCC), Vero E6 cells (a kind gift from C Drosten, Charité Berlin), and TZM-bl cells (obtained from the NIH AIDS Reagent Program) were cultured in DMEM with 10% FBS, 100 µg/ml streptomycin, and 2 mM L-glutamine. For an agarose-overlay, pre-warmed DMEM with 2% FBS mixed with liquid SeaPlaque Agarose (Lonza) to a final concentration of 0.8% agarose was added to cells 2 h postinfection. For plaque assays, Vero E6 cells were overlaid with 2.5% Avicel (Merck) after 1 h of virus inoculation. HEK293T cells expressing vector or IFITM-HA proteins were generated via retroviral transduction and subsequent puromycin selection. Interferon stimulation was performed using indicated concentrations of Roferon (Interferon-$\alpha$2a; Roche).

### Gene editing

*IFITM3*-edited HeLa clones were generated by electroporation of pMAX-CRISPR plasmids encoding EF1$\alpha$ promotor-driven Cas9-2A-EGFP and U6 promoter-driven chimeric gRNAs via the Neon Transfection System (Thermo Fisher Scientific), settings were as recommended by the manufacturer (Neon cell line database, Hela cells): $5 \times 10^6$ cells/ml, 1,005 V, 35 ms pulse width, two pulses. 50 µg/ml of each plasmid was used, for generation of *rs12252* point mutation via HDR, 5 µM ssDNA repair template (Integrated DNA Technologies) was added. After electroporation, EGFP-expressing cells were FACS-sorted and single cell clones were obtained by

limiting dilution. For gRNA target sequences and HDR-template sequence see Table 1. PCR-amplified gene loci of individual clones were genotyped by Sanger sequencing (SeqLab) using the following primers. IFITM3 locus forward: TTTGTTCCGCCCTCATCTGG; IFITM3 locus reverse1 (KO, rs12252. Δ$1^{st}$ ATG): CACCCTCTGAGCATTCCCTG, IFITM3 locus reverse2 (Δexon1): GTGCCAGTCTGGAAGGTGAA, IFITM2 locus forward: CCCTGGCCAGCTCTGCA and IFITM2 locus reverse: CCCCTG GATTTCCGCCAG.

### Plasmids, retro-, and lentiviral vectors

Individual pQCXIP constructs encoding IFITM1-HA, IFITM2-HA, and IFITM-3-HA were obtained by Stephen Elledge (Brass et al, 2009). pQCXIP-IFITM3-HA rs12252, Δ1-21, and $Y^{20}A$ were generated by cloning corresponding gblocks (Integrated DNA Technologies) into pQXCIP using the *NotI* and *AgeI* restriction sites. Retroviral particles were generated by lipofection of HEK293T cells with pQCXIP-IFITM-HA constructs, and plasmids encoding MLV gag pol (Bartosch et al, 2003) and pCMV-VSV-G (Stewart et al, 2003). For production of lentiviral pseudotypes expressing luciferase, HEK293T cells were lipofected with pCSII-EF-luciferase (Agarwal et al, 2006), pCMV DR8.91 (Zufferey et al, 1997) and a plasmid encoding indicated viral glycoprotein, MLV gp and Ebola gp (Chan et al, 2000). CHIKV glycoprotein mutations were introduced into the pIRES2-EGFP-CHIKV E3-E1 (Weber et al, 2017), S27 isolate-based plasmid via site-directed mutagenesis using the QuikChange II Site-Directed Mutagenesis Kit (Agilent Technologies). The following mutations were introduced: E1($A^{226}V$); E1($A^{226}V/M^{269}V$/), E2($K^{252}Q$): sl1; E1($K^{211}N/A^{226}V$), E2($V^{222}I$): sl2; E1($A^{226}V/M^{269}V$), E3($S^{18}F$): sl3A; E1($A^{226}V/M^{269}V$), E2($R^{198}Q$), E3($S^{18}F$): sl3B; E1($A^{226}V/M^{269}V$), E2($L^{210}Q$): sl4. pCHIKrep1 EGFP, encoding CHIKV nonstructural proteins 1–4 was kindly provided by Gorben Pijlman (Fros et al, 2010).

### Viruses

EGFP-CHIKV and EGFP-MAYV were produced by electroporation of in vitro transcribed RNA derived from the molecular clones pCHIK-LR2006-OPY-5'EGFP (Tsetsarkin et al, 2007) and pACYC-MAYV-EGFP (Li et al, 2019b), respectively, into BHK-21 cells. 2 d later, supernatant was filtered, and viral titers were determined by titration on HEK293T cells. In vitro transcribed full-length CHIKV mRNA and mRNA encoding single viral proteins was transfected into target cells with the *Trans*IT mRNA kit (Mirus). MAYV (strain TRVL15537) was passaged on Vero cells. HSV-1 stocks were prepared as previously described (Grosche et al, 2019). Briefly, almost confluent BHK cells were infected at an MOI of 0.01 PFU/cell for 3 d with the BAC-derived strain HSV1(17$^+$)Lox-$_{pMCMV}$GFP, which expresses GFP under the control of the major immediate-early promoter of murine cytomegalovirus (Snijder et al, 2012). The culture medium was collected, cells and debris were sedimented, and HSV-1 particles were harvested by high-speed centrifugation. The resulting virus pellets were resuspended in MNT buffer. Single-use stocks were aliquoted and kept at –80°C. IAV (strain A/PuertoRico/8/34 H1N1) was generated by an eight-plasmid rescue system kindly provided by Richard Webby (St. Jude Children's Research Hospital), using transfection of HEK293T cells and subsequent infection of MDCK cells to generate viral progeny (Hoffmann et al, 2002). HIV clone

pBR-NL4.3-EGFP (Wildum et al, 2006) was transfected into target cells using Lipofectamine 2000 (Thermo Fisher Scientific) and viral titers were determined by titration on Tzm-bl cells.

## Immunoblotting

Cells were lysed with M-PER Mammalian Protein Extraction Reagent (Pierce) and processed according to the recommended protocol. The lysate was mixed with Laemmli buffer and boiled for 10 min at 95°C. Proteins were run on a 10% SDS–PAGE and immobilized on a nitrocellulose membrane (GE Healthcare) using the Trans-Blot Turbo system (Bio-Rad). Blocked membranes were incubated with the following antibodies: mouse anti-MAPK (clone D2, 1:1,000; Santa Cruz Technologies), rabbit anti-Tubulin (2144, 1:1,000; Cell Signaling Technologies), mouse anti-HA (clone 16B12, 1:1,000; BioLegend), mouse anti-IFITM1 (clone 5B5E2, 1:5,000; Proteintech), mouse anti-IFITM2 (clone 3D5F7, 1:5,000; Proteintech), rabbit anti-IFITM3 (Cat. no. AP1153a, 1:500; Abcepta), rabbit anti-CHIKV antiserum (1:10,000; IBT Bioservices), or mouse anti-HIV-1 p24 (1:1,000; ExBio). Secondary antibodies conjugated to Alexa 680/800 fluorescent dyes were used for detection and quantification by Odyssey Infrared Imaging System (LI-COR Biosciences).

## Flow cytometry

Cells were fixed with 4% PFA (Carl Roth) and permeabilized in 0.1% Triton X-100 (Thermo Fisher Scientific) in PBS before immunostaining, if not stated otherwise. Cells were immunostained with the following antibodies: rabbit anti-IFITM3 (Cat. no. AP1153a, 1:500; Abcepta), mouse anti-ISG15 (clone F-9, 1:500; Santa Cruz Technologies), rabbit anti-MX2 (sc-166412, 1:500; Santa Cruz Technologies), mouse anti-BST2 BV421 (566382, 1:40; BD Biosciences) and mouse anti-HA (clone 16B12, 1:1,400; BioLegend). Secondary antibodies conjugated to Alexa Fluor 488 or 647 (1:1,500; Invitrogen) were used for detection. Flow cytometry analysis was performed using FACS Calibur, FACS Celesta, or FACS Accuri with BD CellQuest Pro 4.0.2 Software (BD Pharmingen) and FlowJo V10 Software (FlowJo).

## Immunofluorescence microscopy

Cells were grown in µ-slide eight wells (Ibidi). Cells were fixed with 4% PFA and permeabilized with 0.5% Triton X-100 in PBS. Immunostaining was performed with primary antibodies for 1 h at room temperature for HA (1:1,000; BioLegend) or rabbit anti-IFITM3 (Cat. no. AP1153a, 1:500; Abcepta) and secondary antibodies conjugated to Alexa Fluor 488 and 647 (1:1,000; Invitrogen) for 1 h at room temperature. Nuclear DNA was stained with 2.5 µg/ml DAPI (Invitrogen) for 5 min at room temperature. Microscopic analysis was performed using a Nikon Ti-E microscope equipped with a Yokogawa CSU-X1 spinning-disc and an Andor DU-888 camera. ImageJ was used to prepare microscopy images and for quantification of signal intensity of the immunostaining.

## Quantitative RT-PCR

Cells were lysed and RNA extracted using the Promega Maxwell 16 with the LEV simplyRNA tissue. cDNA was prepared using dNTPs (Thermo Fisher Scientific), random hexamers (Jena Bioscience) and M-MuLV reverse transcriptase (NEB) with buffer. Quantification of relative cellular mRNA levels was performed with the 7500 Fast Real-Time PCR System (Applied Biosystems) or the LightCycler 480 PCR System (Roche) in technical triplicates using Taq-Man PCR technology with the following Taqman probes and primers (Thermo Fisher Scientific): human *IFITM3* (assay ID Hs03057129_s1), *IFIT1* (assay ID Hs01911452_s1), and *RNASEP* (#4316849). Influenza virus RNA replication was assessed by quantifying HA mRNA levels with forward primer CAGATGCAGACACAATATGT and reverse primer TAGTG GGGCTATTCCTTTTA. Relative expression was calculated with the ΔΔCT method, using *RNASEP* or *GAPDH* mRNA as reference.

## Data presentation and statistical analysis

If not otherwise stated, bars and symbols show the arithmetic mean of indicated amount of repetitions. Error bars indicate SD from at least three or SEM from the indicated amount of individual experiments. Statistical analysis was performed with GraphPad Prism 8.3.0 using two-tailed unpaired $t$ tests. $P$-values < 0.05 were considered significant (*), <0.01 very significant (**), <0.001 extremely significant (***); n.s., not significant (≥0.05).

# Supplementary Information

# Acknowledgements

We thank Oliver Dittrich-Breiholz and the Transcriptomics Facility from Hanover Medical School, and Victor Tarabykin for granting access to the Step One Plus Real Time PCR System and the ABI7500 Real Time PCR System, respectively. We thank Hildegard Schilling for technical support with culture and typing of gene-edited cell lines. We thank Bo Zhang for providing the MAYV-GFP clone. We thank the NIH AIDS Research and Reference Reagent Program for providing essential reagents. We thank Thomas Pietschmann and Christian Drosten for constant support. This work was supported by funding from the Deutsche Forschungsgemeinschaft (DFG) for Germany's Excellence Strategy, EXC 2155, project number 390874280 and SFB 900–158989968 (TPC2) awarded to B Sodeik; by grant GO2153/3-1 to C Goffinet within DFG German/African Cooperation Projects in Infectiology, and by DFG grant GO2153/6-1 to C Goffinet, by the Impulse and Networking Fund of the Helmholtz Association through the HGF-EU partnering grant PIE-008 to C Goffinet, and funding of the Helmholtz Center for Infection Research (HZI) and of Berlin Institute of Health (BIH) to C Goffinet.

## Author Contributions

S Franz: investigation.
F Pott: investigation.
T Zillinger: investigation.
C Schüler: investigation.
S Dapa: investigation.
C Fischer: investigation.
V Passos: investigation.
S Stenzel: investigation.

F Chen: investigation.
K Döhner: investigation.
G Hartmann: resources.
B Sodeik: resources.
F Pessler: resources.
G Simmons: resources.
JF Drexler: resources.
C Goffinet: conceptualization.

## Conflict of Interest Statement

The authors declare that they have no conflict of interest.

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
