## [Reviewer comments · Life Science Alliance]

Life Science Alliance

Human IFITM3 restricts Chikungunya and Mayaro virus infection and is susceptible to viral antagonism

Sergej Franz, Fabian Pott, Thomas Zillinger, Christiane Schüler, Sandra Dapa, Carlo Fischer, Vânia Passos, Saskia Stenzel, Fangfang Chen, Katinka Döhner, Gunther Hartmann, Beate Sodeik, Frank Pessler, Graham Simmons, Jan Drexler, and Christine Goffinet

DOI: <https://doi.org/10.26508/lsa.20200909>

Corresponding author(s): *Christine Goffinet, Charité - Universitätsmedizin Berlin*

Review Timeline:

Submission Date:	2020-09-16
Editorial Decision:	2020-11-15
Revision Received:	2021-02-28
Editorial Decision:	2021-03-26
Revision Received:	2021-05-21
Accepted:	2021-05-21

Scientific Editor: Shachi Bhatt

Transaction Report:

November 15, 2020

Re: Life Science Alliance manuscript #LSA-2020-00909-T

Prof. Christine Goffinet
Charité - Universitätsmedizin Berlin
Charitéplatz 1
Berlin 10117
Germany

Dear Dr. Goffinet,

Thank you for submitting your manuscript entitled "Human IFITM3 restricts Chikungunya virus and Mayaro virus infection and is susceptible to virus-mediated counteraction" to Life Science Alliance. The manuscript was assessed by expert reviewers, whose comments are appended to this letter.

As you will note from the reviewers' comments, the reviewers are quite enthusiastic about the findings, but have raised some concerns that should be addressed before further consideration at LSA. While R1 has only requested minor edits, R2 asked about IFITM proteins' expression and some important controls. We encourage you to submit a revised version addressing all the points raised by the reviewers for publication at LSA.

Thank you for this interesting contribution to Life Science Alliance. We are looking forward to receiving your revised manuscript.

Sincerely,

Shachi Bhatt, Ph.D.
Executive Editor
Life Science Alliance
<https://www.lsjournal.org/>
Tweet @SciBhatt @LSAJournal

- A letter addressing the reviewers' comments point by point.
- An editable version of the final text (.DOC or .DOCX) is needed for copyediting (no PDFs).
- High-resolution figure, supplementary figure and video files uploaded as individual files: See our detailed guidelines for preparing your production-ready images, <https://www.life-science-alliance.org/authors>
- Summary blurb (enter in submission system): A short text summarizing in a single sentence the study (max. 200 characters including spaces). This text is used in conjunction with the titles of papers, hence should be informative and complementary to the title and running title. It should describe the context and significance of the findings for a general readership; it should be written in the present tense and refer to the work in the third person. Author names should not be mentioned.

B. MANUSCRIPT ORGANIZATION AND FORMATTING:

Reviewer #1 (Comments to the Authors (Required)):

This paper provides strong evidence that IFITM antiviral proteins restrict infection by Chikungunya virus. This conclusion is bolstered by the use of both loss of function (CRISPR KO) and gain of function (over expression) approaches. Perhaps the most exciting observation in this paper is that CHIKV decreases IFITM protein levels in infected cells, a finding that was recapitulated by transfecting the virus genome or the viral non-structural proteins into cells. This suggests that CHIKV targets IFITMs as a countermeasure to their antiviral effects. While the authors have narrowed this effect on IFITM3 down to the non-structural proteins, they have not identified a

specific viral protein that performs this function or a mechanism by which this occurs. However, the general observation is of high importance and provides a rare example of a virus that targets IFITMs. The authors also provide a reasonable discussion speculating that downregulation of IFITMs in infected cells may prevent IFITM incorporation into progeny virions, which has been shown to decrease infectivity of other viruses. Overall, this study provides important advances in describing that IFITMs block CHIKV infection and that CHIKV in turn encodes a mechanism to decrease IFITM expression. While I am overall supportive of publication of this work, a few minor issues should be addressed as listed below.

1. Representative raw data (flow cytometry plots) for infection experiments should be provided.
2. As judged by flow cytometry histograms, all three rs12252-C HeLa clones expressed lower baseline levels of IFITM3 than the parental cell line both before and after IFN treatment. The authors should comment on this rather than saying the level of expression is similar to the parental line. Is it possible that the SNP affects IFITM3 expression (albeit to a moderate degree that does not seem to affect virus infection of HeLa cells)?
3. I would suggest removing Mayaro virus from the title as there is minimal data provided for this virus.

Reviewer #2 (Comments to the Authors (Required)):

Interferon induced transmembrane (IFITM) proteins have been shown to inhibit the replication of a wide range of enveloped viruses. Although IFITM restriction of alphaviruses has been described in published work, detailed mechanistic studies have been limited to Semliki Forest Virus (SFV), an alphavirus that is not, for the most part, pathogenic in man. In this paper the authors investigate the ability of human IFITM proteins to restrict infection by two human alphavirus pathogens, Chikungunya virus (CHIKV) and Mayaro virus (MAYV), in human cell systems. The authors show that both viruses are restricted by all three human IFITMs, with IFITM3 being the most potent, and that restriction is mediated at the level of entry. However, the authors also report a novel and potentially interesting finding that CHIKV infection leads to a posttranscription decrease in IFITM3 expression that appears to be mediated through one or more of the viral non-structural proteins. While Figs 1-3 extend previous work, showing IFITM protein restriction of alphaviruses, the data presented in Figs 4 and 5, relating to the decreased expression of IFITM proteins in CHIKV infected cells, is novel but its relevance is not investigated in the current study.

Overall, the paper adds to understanding of IFITM protein-mediated viral restriction, however there are a number of points the authors should consider/address before the paper is acceptable for publication. Addressing the points below may require additional experiments, but it should be possible to do these in a few weeks.

In the experiments illustrated in Figure 2; exogenous IFITM constructs are expressed. How do the levels of expression of each IFITM protein compare to the levels of endogenous protein expressed with and without IFN treatment? Could the apparent restriction of CHIKV by IFITM1 be due to overexpression of IFITM1? Do the authors have any idea why Y20A and Δ 1-21 (IFITM3 mutants expressed on the cell surface) fail to restrict virus, but IFITM1, which is primarily located at the cell surface, does restrict?

In Figure 4, the analysis of IFITM protein expression seems to be restricted to cell surface protein, and does not consider total cellular proteins, why is this? Total cell IFITM protein should be reported, preferably by western blot. Why does IFITM2 appear to be on the cell surface in Fig 4C; this protein is usually seen primarily in endosomes/lysosomes?

In the experiments showing an apparent down regulation of IFITM3 in CHIKV infected cells, MX1 and ISG15 are used as controls to argue against global effects on translation: it would be appropriate to also look at membrane proteins? Is the same effect seen in Mayaro virus infected cells? These experiments raise a number of questions. For example, how does down regulation of IFITM3 in an infected cell benefit the virus? Published work with HIV has demonstrated the incorporation of IFITM proteins into virions reduces particle infectivity. One could imagine that a similar scenario might apply for CHIKV. However, alphaviruses assemble differently and may not package IFITM proteins. Can the authors demonstrate that CHIKV-induced down modulation has some positive impact on CHIKV transmission/replication?

Minor points:-

Line 35/36: not strictly true; Weston et al. used human cells in their studies.

Line 66: delete 'among others'

Line 144: Fig 1F should be Fig 1E

Line 157-161, Fig 2A: the authors should explain the double IFITM2 band seen with anti-IFITM2 antibodies (only one band is seen for IFITM2 with anti-HA). Given the FACS analysis (Fig 2B) is done with anti-HA, is the total amount of IFITM2 underestimated in these analyses?

Line 212: 'punctuated' should be 'punctate'

Reviewer #1 (Comments to the Authors (Required)):

This paper provides strong evidence that IFITM antiviral proteins restrict infection by Chikungunya virus. This conclusion is bolstered by the use of both loss of function (CRISPR KO) and gain of function (over expression) approaches. Perhaps the most exciting observation in this paper is that CHIKV decreases IFITM protein levels in infected cells, a finding that was recapitulated by transfecting the virus genome or the viral non-structural proteins into cells. This suggests that CHIKV targets IFITMs as a countermeasure to their antiviral effects. While the authors have narrowed this effect on IFITM3 down to the non-structural proteins, they have not identified a specific viral protein that performs this function or a mechanism by which this occurs. However, the general observation is of high importance and provides a rare example of a virus that targets IFITMs. The authors also provide a reasonable discussion speculating that downregulation of IFITMs in infected cells may prevent IFITM incorporation into progeny virions, which has been shown to decrease infectivity of other viruses. Overall, this study provides important advances in describing that IFITMs block CHIKV infection and that CHIKV in turn encodes a mechanism to decrease IFITM expression. While I am overall supportive of publication of this work, a few minor issues should be addressed as listed below.

Reply: Thank you very much for the overall positive comments and your constructive review.

1. Representative raw data (flow cytometry plots) for infection experiments should be provided.

Reply: We have now provided supplemental figures containing representative dot plots of several experiments (Fig. S2 for Fig. 1D and E; Fig. S3 for Fig. 2D; Fig. S4 for Fig. 3A and C; Fig. S5 for Fig. 4E; Fig. S6 for Fig. 5C).

2. As judged by flow cytometry histograms, all three rs12252-C HeLa clones expressed lower baseline levels of IFITM3 than the parental cell line both before and after IFN treatment. The authors should comment on this rather than saying the level of expression is similar to the parental line. Is it possible that the SNP affects IFITM3 expression (albeit to a moderate degree that does not seem to affect virus infection of HeLa cells)?

Reply: thanks for pointing this out. Indeed, the levels of IFITM3 rs12252-C are lower than for wild-type IFITM3 in the flow cytometric analysis of permeabilized cells, and this difference is also detectable in the immunoblot. We rephrased the corresponding text passages as following:

“In contrast, three clones expressing the rs12252 T-to-C variant displayed detectable IFN-induced IFITM3 expression, although slightly lower than in parental cells.” (page 6, lines 129-131)

“Flow cytometry analysis of IFITM3 protein expression in permeabilized cells paralleled the results of the immunoblot analysis, with IFITM3 in rs12252-C cells being slightly less abundant at baseline and upon IFN stimulation (Figure 1B) (pages 6-7, lines 134-137)

“However, IFITM-3 rs12252-C seemed to be expressed to slightly lower levels than wild-type IFITM3, without impacting its antiviral efficacy.” (page 14, lines 324-325)

3. I would suggest removing Mayaro virus from the title as there is minimal data provided for this virus.

Reply: In the revised version, we are now adding new datasets for MAYV addressing the role of endogenous IFITM3 during MAYV infection (Fig. 1E) and the MAYV-mediated counteraction of IFITM3 (Fig. 5 C-D). We therefore feel it appropriate to leave MAYV in the title.

Reviewer #2 (Comments to the Authors (Required)):

Interferon induced transmembrane (IFITM) proteins have been shown to inhibit the replication of a wide range of enveloped viruses. Although IFITM restriction of alphaviruses has been described in published work, detailed mechanistic studies have been limited to Semliki Forest Virus (SFV), an alphavirus that is not, for the most part, pathogenic in man. In this paper the authors investigate the ability of human IFITM proteins to restrict infection by two human alphavirus pathogens, Chikungunya virus (CHIKV) and Mayaro virus (MAYV), in human cell systems. The authors show that both viruses are restricted by all three human IFITMs, with IFITM3 being the most potent, and that restriction is mediated at the level of entry. However, the authors also report a novel and potentially interesting finding that CHIKV infection leads to a posttranscription decrease in IFITM3 expression that appears to be mediated through one or more of the viral non-structural proteins. While Figs 1-3 extend previous work, showing IFITM protein restriction of alphaviruses, the data presented in Figs 4 and 5, relating to the decreased expression of IFITM proteins in CHICK infected cells, is novel but its relevance is not investigated in the current study.

Overall, the paper adds to understanding of IFITM protein-mediated viral restriction, however there are a number of points the authors should consider/address before the paper is acceptable for publication. Addressing the points below may require additional experiments, but it should be possible to do these in a few weeks.

Reply: Thank you very much for the overall positive comments and your constructive review.

In the experiments illustrated in Figure 2; exogenous IFITM constructs are expressed. How do the levels of expression of each IFITM protein compare to the levels of endogenous protein expressed with and without IFN treatment? Could the apparent restriction of CHIKV by IFITM1 be due to overexpression of IFITM1? Do the authors have any idea why Y20A and Δ 1-21 (IFITM3 mutants expressed on the cell surface) fail to restrict virus, but IFITM1, which is primarily located at the cell surface, does restrict?

Reply: Thank you for this comment. Quantification of protein expression in the endogenous (HeLa) and heterologous (HEK293T) expression context revealed that overexpression reaches protein expression levels that we obtained by IFN treatment (date not included in the manuscript).

[Figure removed by editorial staff per authors' request].

We included the following text passage to address this limitation:

“Heterologous expression of genes can cause aberrant subcellular localization and/or non-physiological expression levels. Therefore, results obtained by heterologous expression need to be interpreted with caution. Importantly, levels of IFITM protein expression obtained by heterologous expression of IFITM were similar to levels induced by IFN treatment of HeLa cells (data not shown). Future studies are warranted to corroborate the contribution of IFITM1 to alphavirus restriction.” (page 15, lines 336-341)

One idea to reconcile the presence of antiviral activity of plasma membrane-localizing IFITM1 and absence of antiviral activity of plasma membrane-localizing IFITM3 mutants could be that IFITM1 exerts an IFITM1-exclusive antiviral activity via a protein motif that is not available in IFITM3. However, we have no data to support this hypothesis and thus would prefer to not include this idea into the discussion.

In Figure 4, the analysis of IFITM protein expression seems to be restricted to cell surface protein, and does not consider total cellular proteins, why is this? Total cell IFITM protein should be reported, preferably by western blot. Why does IFITM2 appear to be on the cell surface in Fig 4C; this protein is usually seen primarily in endosomes/lysosomes?

Reply: We conducted quantitative immunoblots aiming to detect potential changes of IFITM expression (new dataset, Fig. 4C). This analysis failed to detect statistically significant reductions of IFITM protein abundance. However, this may be related to the fact that we were unable to successfully conduct immunoblotting on cells infected with a sufficiently high MOI (due to cytopathic effect), and we had to restrict our analysis to MOI 0.1 and 1. So, a potential degradation of IFITM proteins might be more challenging to detect in this bulk analysis. Indeed, our inspection of single GFP cells by immunofluorescence argue for a reduction of steady-state levels of antiviral IFITMs. We added the following text:

“While quantitative immunoblotting of whole cell lysates failed to detect a reduction of IFITMs upon infection (Fig. 4C), single cell inspection of EGFP-positive cells by immunofluorescence suggested that antivirally active IFITMs were reduced in quantity

upon infection (Fig. 4D).” (page 11, lines 233-236)

In the experiments showing an apparent down regulation of IFITM3 in CHIKV infected cells, MX1 and ISG15 are used as controls to argue against global effects on translation: it would be appropriate to also look at membrane proteins? Is the same effect seen in Mayaro virus infected cells? These experiments raise a number of questions. For example, how does down regulation of IFITM3 in an infected cell benefit the virus? Published work with HIV has demonstrated the incorporation of IFITM proteins into virions reduces particle infectivity. One could imagine that a similar scenario might apply for CHIKV. However, alphaviruses assemble differently and may not package IFITM proteins. Can the authors demonstrate that CHIKV-induced down modulation has some positive impact on CHIKV transmission/replication?

Reply: Thank you for these questions. We have now added new data addressing these questions. First, IFITM3 downregulation was also detectable for MAYV-infected cells (Fig. 5C, Fig. S6). Interestingly, CD317/tetherin was also downregulated by CHIKV infection, suggesting a counteraction strategy against this second restriction factor (Fig. 5C, Fig. S6). Finally, we addressed a potential ability of IFITM3 to mediate imprinting of CHIKV and are now devoting an entire new main figure (Fig. 6) to this question. Excitingly, our new datasets provides evidence for IFITM3-mediated reduction of particle infectivity of CHIKV, providing a rationale for the necessity of a virus-encoded counteraction strategy against this restriction factor. The following text was added:

“Reduction of particle infectivity through expression of endogenous IFITM3 in alphavirus-producing cells. Reduction of IFITM3 protein abundance upon alphaviral infection suggested that IFITM3 exerted antiviral functions beyond entry inhibition. Therefore, we determined the impact of IFITM3 expression in virus-producing cells on the infectivity of progeny virions. IFITM3 can incorporate into viral particles and reduce their ability to infect new cells (Appourchaux, Delpeuch et al., 2019, Compton et al., 2014, Tartour, Appourchaux et al., 2014). To bypass the impact of IFITM proteins on virus entry, we transfected cells expressing IFITM3 endogenously or exogenously, and their IFITM3-negative counterparts. 24 hours post transfection, we analyzed the abundance of viral capsid and amount of infectivity in the supernatant (Fig. 6A). Capsid abundance was lower in supernatants from IFITM3-expressing cells, which however was related to a generally lower transfection rate and lower cell-associated capsid expression. We failed to detect virion-associated IFITM3 in the supernatant of any infected cell cultures under these experimental conditions (Fig. 6A). Viral titers, when normalized for differences in transfection efficiencies, was not significantly influenced by the IFITM3 expression status (Fig. 6B). However, the specific infectivity, defined as the infectivity per capsid, was two-fold reduced for CHIKV and HIV-1 produced in IFITM3-expressing parental HeLa cells and for HIV-1 produced in HEK293T cells expressing IFITM3-HA. However, the specific infectivity of CHIKV did not seem to be affected by heterologous IFITM3 expression. These data provide first evidence for a potential ability of endogenous IFITM3 to negatively imprint nascent CHIKV, in addition to inhibition of virus entry. Future work is required to establish the relative contribution of these two IFITM3-mediated antiviral strategies in the context of alphaviral infection.” (pages 12-13, lines 270-321).

Minor points:

Line 35/36: not strictly true; Weston et al. used human cells in their studies.

Reply: Thank you. We have rephrased the sentence as following:

“The role of IFITM proteins during alphaviral infection of human cells and viral counteraction strategies are insufficiently understood” (page 2, line 36)

Line 66: delete 'among others'

Reply: Thank you. We have deleted “among others”.

Line 144: Fig 1F should be Fig 1E

Reply: Thank you. We have corrected the figure assignment.

Line 157-161, Fig 2A: the authors should explain the double IFITM2 band seen with anti-IFITM2 antibodies (only one band is seen for IFITM2 with anti-HA). Given the FACS analysis (Fig 2B) is done with anti-HA, is the total amount of IFITM2 underestimated in these analyses?

Reply: Thank you. We have been unable to solve this issue. We have repeatedly sequenced our plasmids to exclude a contamination. Puzzingly, we do not always detect the double band (see Fig. 4C). However, the phenotypes upon transfection of the IFITM2-HA were stable.

Line 212: 'punctuated' should be 'punctate'

Reply: Thank you. We have corrected this typo.

March 26, 2021

RE: Life Science Alliance Manuscript #LSA-2020-00909-TR

Prof. Christine Goffinet
Charité - Universitätsmedizin Berlin
Charitéplatz 1
Berlin, Berlin 10117
Germany

Dear Dr. Goffinet,

Thank you for submitting your revised manuscript entitled "Human IFITM3 restricts Chikungunya and Mayaro virus infection and is susceptible to viral antagonism". We would be happy to publish your paper in Life Science Alliance pending final revisions necessary to address the remaining minor concern of Reviewer 1 and meet our formatting guidelines.

Along with the points listed below, please also attend to the following,

- please add Keywords, Category, and Summary Blurb/Alternate Abstract for your manuscript in our system
- please be sure that you added Author Contributions of all Authors to your main manuscript text
- please upload your Table in editable .doc or excel format
- please add a callout for Figure 6C to your main manuscript text
- please add your table legend to the main manuscript text after the main and supplementary figure legends
- please use the [10 author names, et al.] format in your references (i.e. limit the author names to the first 10)
- please provide the unedited source image for Figure 5A row 1 panel 3 and Figure 4C vector column, row 1
- please provide better images for blots shown in Figure 1A row 3 (anti-IFITM2) and Figure 6A, HeLa cells treated with supernatant and blotted for anti-HIV1 p24 capsid and anti-HA/anti-IFITM3

A. FINAL FILES:

B. MANUSCRIPT ORGANIZATION AND FORMATTING:

Sincerely,

Shachi Bhatt, Ph.D.
Executive Editor

Life Science Alliance

<https://www.lsjournal.org/>

Interested in an editorial career? EMBO Solutions is hiring a Scientific Editor to join the international Life Science Alliance team. Find out more here -

https://www.embo.org/documents/jobs/Vacancy_Notice_Scientific_editor_LSA.pdf

Reviewer #1 (Comments to the Authors (Required)):

The authors have addressed my previous concerns. I also appreciate the addition of important new data in Figure 6. However, statistical analysis should be added to to this figure.

May 21, 2021

RE: Life Science Alliance Manuscript #LSA-2020-00909-TRR

Prof. Christine Goffinet
Charité - Universitätsmedizin Berlin
Charitéplatz 1
Berlin, Berlin 10117
Germany

Dear Dr. Goffinet,

Thank you for submitting your Research Article entitled "Human IFITM3 restricts Chikungunya and Mayaro virus infection and is susceptible to viral antagonism". It is a pleasure to let you know that your manuscript is now accepted for publication in Life Science Alliance. Congratulations on this interesting work.

DISTRIBUTION OF MATERIALS:

Again, congratulations on a very nice paper. I hope you found the review process to be constructive and are pleased with how the manuscript was handled editorially. We look forward to future exciting submissions from your lab.

Sincerely,

Shachi Bhatt, Ph.D.

Executive Editor

Life Science Alliance

<http://www.lsjournal.org>
